# Tissue-Level Effect of Andrographis and Ashwagandha Metabolites on Metabolic and Inflammatory Gene Expression in Skeletal Muscle and Adipose Tissue: An Ex Vivo/In Vitro Investigation

**DOI:** 10.3390/nu16142291

**Published:** 2024-07-17

**Authors:** Celeste Lugtmeijer, Joanna L. Bowtell, Mary O’Leary

**Affiliations:** Faculty of Health and Life Sciences, Department of Public Health and Sport Sciences, University of Exeter, Exeter EX1 2LU, UK; cl898@exeter.ac.uk (C.L.); j.bowtell@exeter.ac.uk (J.L.B.)

**Keywords:** ashwagandha, *Withania somnifera*, Andrographis, *Andrographis paniculata*, anti-inflammatory, antioxidant, obesity, adipose tissue, skeletal muscle, ex vivo

## Abstract

Adipose tissue and skeletal muscle dysfunction play a central role in cardiometabolic morbidity. Ashwagandha and Andrographis are purported to have anti-inflammatory and antioxidant activity, but this is based on exposure of cells to the parent compounds ignoring phytochemical absorption and metabolism. We explored the anti-inflammatory/antioxidant effects of ashwagandha and Andrographis in ex vivo human models of skeletal muscle and adipose tissue. Healthy participants supplemented with 2000 mg/day Andrographis (n = 10) or 1100 mg/day ashwagandha (n = 10) for 28 days. Sera collected pre (D0) and post (D28) supplementation were pooled by timepoint and added to adipose explant (AT) and primary human myotube (SKMC) culture media (15% *v*/*v*) for treatment. A Taqman panel of 56 genes was used to quantify these. In AT, treatment with ashwagandha sera decreased the expression of genes involved in antioxidant defence and inflammatory response (*CCL5*, *CD36*, *IL6*, *IL10*, *ADIPOQ*, *NFEL2*, *UCP2*, *GPX3*, *GPX4*; geometric 95% CI for fold change > 1) and altered the expression of genes involved in fatty acid metabolism. In SKMC, ashwagandha sera altered *FOXO1* and *SREBF1* expression. Andrographis sera decreased *IL18* and *SERPINEA3* expression in AT. This physiologically relevant in vitro screening characterises the effects of ashwagandha in AT to guide future clinical trials.

## 1. Introduction

Obesity has risen to epidemic levels worldwide generating a significant global health burden. Increased adiposity is associated with chronic diseases including diabetes, hypertension, hyperlipidaemia, and other cardiovascular diseases [1]. Adipose tissue (AT) milieu is complex and dynamic and changes with energy balance. AT expansion through adipocyte hypertrophy and hyperplasia results in immune cell infiltration of AT and a switch from ‘classically activated’ (M2)- to ‘alternatively activated’ (M1)-type macrophages, triggering the release of pro-inflammatory cytokines. This process leads to adipocyte hypoxic necrosis and adipose tissue dysfunction [2,3,4,5]. Ultimately, this results in an impaired ability to store excess triglycerides, and lipids accumulate in the systemic circulation [6]. Ectopic fat deposition occurs in skeletal muscle (SKM), liver, and pancreas [7]. In SKM, this leads to impaired fatty acid metabolism, reduced insulin-mediated glucose uptake, and the formation of toxic lipid intermediates [8]. Tissue-level energy metabolism dysfunctions further promote the systemic and tissue-level inflammation and the insulin resistance that accompany the consequences of obesity (e.g., type 2 diabetes mellitus; T2D) [9,10].

Finding new strategies to prevent the consequences of obesity is a major public health challenge. Additional approaches are needed to complement traditional lifestyle advice. In vivo and pre-clinical evidence suggests that supplementation with certain herbal supplements could improve metabolic health and dampen inflammation at the systemic and tissue level [11]. A limited number of studies have investigated the effects of ashwagandha and Andrographis on inflammation and metabolic dysfunction in AT and peripheral tissues (e.g., SKM). 

Ashwagandha (*Withania somnifera*) root and its bioactive compound Withaferin A have been suggested to have anti-obesity effects [12,13]. Withaferin A treatment (1–25 μM) has been shown to exert anti-adipogenic and pro-apoptotic activity in 3T3-L1 adipocytes [14]. In pre-clinical studies, ashwagandha supplementation leads to significant weight loss [15,16]. In mice, supplementation with Withaferin A or ashwagandha extract reduced lipid accumulation in AT and promoted the browning of subcutaneous AT via increasing the expression of mitochondrial uncoupling protein 1 (UCP-1) [17,18]. Additionally, ashwagandha supplementation improved mitochondrial function (upregulated UCP2 and peroxisome proliferator-activated receptor gamma coactivator 1-alpha (PGC1α) mRNA expression) in mice and increased the oxygen consumption rate in C2C12 mouse myoblasts [17]. 

Andrographis (*Andrographis paniculate*, Burm. F) leaf and its main bioactive ingredient andrographolide have also been found to promote cardiometabolic health [19,20]. Andrographolide treatment dose-dependently (0–15 μM) inhibited the differentiation of 3T3-L1 preadipocytes via the inhibition of CCAAT-enhancer-binding proteins (C/EBP)α and C/EBPβ mRNA and protein expression [21]. Andrographolide also inhibited adipogenesis via the inhibition of peroxisome proliferator-activated receptor gamma (PPARγ) and blocked the gene expression of PPARγ-targeted genes [21,22]. Adipogenesis and the differentiation of preadipocytes into mature adipocytes are also promoted by increased endogenous reactive oxygen species (ROS) [23]. Andrographolide inhibited ROS production in preadipocytes, further inhibiting preadipocyte differentiation [24]. In mice with muscular dystrophy, 3 months of intraperitoneal injection andrographolide 1.0 mg/kg/day significantly reduced nuclear factor kappa B (NF-κB) activity in SKM. Investigations in other tissues support the anti-inflammatory effect of Andrographis/andrographolide [25,26,27,28,29]. However, the concentrations used exceed those found in vivo, and in vivo studies used intraperitoneal injections rather than oral administration, thus bypassing the gastrointestinal (GI) tract and ignoring any challenges with bioavailability. 

The existing literature investigating the effects of ashwagandha and Andrographis on metabolism and inflammation consists of non-human (murine and in vitro) investigations and focuses on the main bioactive compound alone. Often, plant extracts (or a selection of their compounds) are applied to cells, and a response is measured. This strategy negates their metabolism in the GI tract, bioavailability, phase 2 metabolism, and interactions between metabolites. Many of these plant compounds have low bioavailability and undergo extensive metabolism before reaching peripheral tissues (e.g., AT, SKM) [30,31]. Applying parent plant compounds at supraphysiological concentrations to tissues is an inadequate predictor of in vivo response. The effects of ashwagandha and Andrographis metabolites on human AT and SKM inflammation, oxidative stress, and metabolism are currently unknown. 

Here, we explored the physiological effects of Andrographis and ashwagandha in physiologically relevant ex vivo human models of SKM and AT. We measured the mRNA expression of 56 genes related to inflammation, pro-oxidant/anti-oxidant balance, and metabolism. We hypothesised that treatment of AT and SKM with the sera of people who supplemented with Andrographis or ashwagandha would inhibit pro-inflammatory genes (AT and SKM), promote the expression of genes involved in the anti-inflammatory/antioxidant defence (AT and SKM), and have anti-adipogenic effects (AT).

## 2. Materials and Methods

### 2.1. Participants

Twenty healthy, lean (body mass index [BMI] < 25 kg·m^−2^), young (age < 40 years old) participants were recruited from the University of Exeter and surrounding areas (Table 1). All participants provided their written informed consent. The study was conducted in accordance with the Declaration of Helsinki, and approved by the Ethics Committee of The University of Exeter (ref: 22-02-02-A-06 [supplementation and blood collection], approval date: 24 February 2022 and the reference 516627 [adipose tissue explant collection], approval date: 16 October 2023). Participants were excluded from the study if they were taking other herbal supplements or had completed a course of antibiotics, corticosteroids, or immunosuppressive treatment in the last 6 months. Participants were instructed to maintain their usual diet and lifestyle over the course of the experiment. Study visits occurred after an overnight fast (>12 h). 

Baseline venous blood samples were collected by venepuncture of an antecubital vein. Participants underwent 28 days (+/− 2 days) of supplementation with either powdered ashwagandha root (n = 10, 1100 mg/day) (Pukka Herbs, Keynsham, UK) or powdered Andrographis leaf (n = 10, 2000 mg/day) (Pukka Herbs, Keynsham, UK) capsules. Empirical evidence regarding ashwagandha doses in humans is limited; this dose was chosen as (1) a similar dose (500 mg twice daily) has been shown to improve endothelial function in those with diabetes mellitus [32] and (2) this dose was in line with the manufacturer’s recommendations. Similarly, few human Andrographis trials exist, with leaf extracts, rather than pure powdered leaf, typically being used [33,34,35]. Therefore, the manufacturer’s recommended dose was used. This is an ecologically valid approach, as it is presumably the dose that is being consumed by existing Andrographis users. Participants’ blood was taken at baseline (BL) and after they returned at day 28. A blood sample was taken immediately upon their arrival at the laboratory (chronic supplementation only (CH)) and 1 h after their final dose of supplement (chronic supplementation + acute supplementation (CA)) (Figure 1). To obtain serum, blood was rested at room temperature for up to 1 h, centrifuged for 15 min at 4500 rpm at 4 °C, and immediately stored at −70 °C until use. Serum was thawed and pooled (1:1) per condition and timepoint. The pooled sera were then stored at −70 °C and thawed at 4 °C for cell culture experiments. The pooled sera were filtered through a 0.22 µm polyethersulfone membrane prior to cell culture applications.

#### Adipose Tissue

Following ethical approval (ref: 516627), AT was obtained from 7 healthy volunteers (Table 2) between March 2023 and April 2024. Volunteers were recruited from the University of Exeter Sports and Health Science department and provided written informed consent. Participants were excluded if they met any of the following criteria: ≥40 years old; BMI ≥ 27 kg·m^−2^; currently taking dietary supplements (except vitamin D); frequent use of medication or recreational drugs likely to affect our results; have a bleeding disorder or taking medication that impairs blood clotting; Hepatitis B, Hepatitis C, or Human Immunodeficiency Virus positive; have had an adverse reaction to a local anaesthetic in the past; have a skin condition that is likely to increase the risk of infection at the biopsy site; pregnant; recent (<2 weeks) infection or vaccination.

Subcutaneous AT was obtained from ~5 cm lateral to the umbilicus with a 14 G needle using the needle aspiration method [36] under local anaesthesia (B Braun, Sheffield, UK). The AT was then washed with phosphate-buffered saline (PBS), strained in a 50 µm mesh cell strainer, and visible clots were removed using a forceps and scalpel. The tissue was transported in basal endothelial cell media (Promocell, Heidelberg, Germany). The AT was weighed and immediately plated in triplicates in a 24-well plate with ~50 mg of tissue per well. The AT was incubated for 24 h in 300 µL/well basal endothelial cell media before treatment. 

### 2.2. Primary Human Myogenic Cell Culture

Primary human myogenic cells (SKMC) were isolated from the vastus lateralis muscle biopsy of a healthy (BMI: 22.2 kg·m^−2^) 19-year-old male from a previous study [37]. This biopsy was collected at baseline (pre-intervention) using the suction-modified percutaneous Bergstrom needle technique [38].

SKMC were cultured as described previously [39]. Briefly, the SKM biopsy was minced and digested on an orbital rotator for 15 min at 37 °C. An amount of 5 mL growth medium (Ham’s F10 with 1 mM L-glutamine, 1% penicillin-streptomycin, 20% FBS) was added and the digest centrifuged at 460× *g* for 5 min. The resulting pellets were resuspended in growth medium in a T25 cell culture flask in a humidified 37 °C incubator with an atmosphere of 5% CO_2_. Myoblasts were subcultured (1:3) at 75% confluence by trypsin-induced dissociation from their vessel and were used experimentally at passages 4–7. Cells were seeded in 24-well plates at 75% confluence in triplicates for each condition.

### 2.3. Treatment

After 24 h in culture in 24-well plates, AT and SKMC were treated with basal endothelial cell culture media supplemented with 15% pooled sera of BL, CH, and CA for 24 h (Figure 2). Treatments were plated in triplicate (3 wells per condition) and in biological quadruplicates (4 independent experiments). After 24 h of treatment, cell culture media were discarded and cells/tissues were immediately lysed with TRizol reagent (Thermo Fisher, Foster City, CA, USA) and stored at −20 °C until RNA extraction. AT lysis was carried out using a bead homogeniser (Speedmill Plus; Analytik Jena AG, Jena, Germany)) in 120 s cycles until no visible tissue remained. SKMC lysis was carried out according to the TRizol manufacturer’s instructions.

### 2.4. OpenArray

TRI reagent (Thermo Fisher, Foster City, CA, USA) was used to isolate RNA according to the manufacturer’s instructions. QuBit Broad range RNA kit (Thermo Fisher, Foster City, CA, USA) with the Qubit 4 Fluorometer (Invitrogen, Waltham, MA, USA) was used to quantify RNA according to the manufacturer’s instructions. 

Total RNA (500 ng) was reverse transcribed in 10 µL reactions using the Superscript III VILO kit (Thermo Fisher, Foster City, CA, USA). An amplification test-plate was run to confirm amplification. Polymerase chain reaction (PCR) reactions for the amplification test plate were run in duplicate and contained 2.5 µL SYBR Green mastermix (Bio-Rad Laboratories, Hercules, CA, USA), 0.25 µL IL-6 primer (GPX1, assay ID qHsaCED0037003) (Bio-Rad Laboratories, Hercules, CA, USA), and 2.25 µL (5 ng) cDNA. PCR conditions were a single hold stage of 95 °C for 30 s, followed by 40 cycles of 95 °C for 10 s and 60 °C for 30 s.

Expression of the 56 selected genes was measured by OpenArray qRT PCR relative to 4 endogenous control genes (*GAPDH*, *TBP*, *ACTB*, *RPLP0*) on the QuantStudio 12K Flex platform (Thermo Fisher, Foster City, CA, USA). PCR reactions were run in triplicates and contained 3.8 µL TaqMan OpenArray Real-Time PCR Master Mix (Thermo Fisher, Foster City, CA, USA), and 1.2 µL cDNA in a total volume of 5 µL. PCR conditions were a single cycle of 95 °C for 10 min followed by 40 cycles of 95 °C for 15 s and 60 °C for 1 min. Expression levels were quantified by the Comparative Ct approach and normalised to expression levels in cells treated with baseline sera.

### 2.5. Data Analysis

The relative expression of each gene was calculated using the 2-deltadeltaCT method in Microsoft Excel (Version 2403). The RNA expression in the control cells/tissue was designated as 1, and the relative levels of the gene transcripts in the samples were expressed as relative expression. Results are presented as geometric means (Gmean) and 95% confidence intervals. Confidence intervals that did not include the value of zero effect (i.e., the reference value of 1) can be assumed to be statistically significant [40]. Because the aim of this study was explorative, the geometric mean and 95% confidence interval were judged more informative than statistical tests to characterise the true response to treatment (rather than a binary yes/no response to the treatment) [41]. 

## 3. Results

### 3.1. Andrographis

Treatment of adipose explants and SKMC with the sera of participants who supplemented with Andrographis for 28 days (chronic supplementation and chronic + acute supplementation) had no effect on the mRNA expression of *LEP*, *ADIPOQ*, *NAMPT*, *RETN*, *CPT1B*, *SLC2A4*, *GSK3A*, *GSK3B*, *PDK4*, *PPARGC1A*, *PPARA*, *PPARD*, *PPARG*, *AKT1*, *CPT2*, *FOXO1*, *PPARG*, *FASN*, *FABP4*, *LPL*, *UCP1*, *UCP2*, *SERPINE1*, *SREBF1*, *CD36*, *IL-6*, *IL1β*, *IL-10*, *CD14*, *NFkB1*, *CCL2*, *MMP9*, *MMP14*, *APOE*, *CCL5*, *IFNG*, *SOD1*, *SOD2*, *GPX1*, *GPX3*, *GPX4*, *CAT*, *NFE2L2*, *COL1A1*, *iNOS*, *UCP2.*


AT treated with AND-CH had significantly lower *IL18* mRNA expression (Gmean: 0.338, 95% CI: 0.132, 0.767) (Figure 3). There was no effect of treatment on *IL18* mRNA expression in SKMC. 

AT treated with the sera of participants who underwent chronic Andrographis supplementation had significantly lower relative mRNA expression of *SERPINA3* (Gmean: 0.224, 95% CI: 0.040, 0.867). There was no effect of treatment on *SERPINA3* mRNA expression in SKMC. 

### 3.2. Ashwagandha

AT treated with ashwagandha metabolites had significantly lower expression of genes involved in fatty acid uptake, fat metabolism, and carbohydrate/glucose metabolism (Figure 4). ASH-CH- and ASH-CA-treated AT had lower expression of ***CD36*** (ASH-CH: 0.272, 95% CI: 0.00, 0.853; ASH-CA: 0.207, 95% CI: 0.072, 0.542), ***CPT1B*** (ASH-CH: 0.254, 95% CI: 0.072, 0.817), ***CPT2*** (ASH-CH: 0.194, 95% CI: 0.00, 0.512; ASH-CA: 0.247, 95% CI: 0.037, 1.317), ***FABP4*** (ASH-CH: 0.384, 95% CI: 0.133, 1.060; ASH-CA: 0.271, 95% CI: 0.063, 1.148), ***ADIPOQ*** (ASH-CH: 0.327, 95% CI: 0.00, 1.019; ASH-CA: 0.247, 95% CI: 0.140, 0.458), ***FASN*** (ASH-CH: 0.40, 95% CI: 0.149, 0.340; ASH-CA: 0.619, 95% CI: 0.258, 1.438), ***NAMPT*** (ASH-CH: 0.240, 95% CI: 0.058, 0.925; ASH-CA: 0.203, 95% CI: 0.081, 0.488), ***GSK3A*** (ASH-CH: 0.307, 95% CI: 0.104, 0.839; ASH-CA: 0.264, 95%CI: 0.128, 0.527), ***CCL5*** (ASH-CH: 0.188, 95% CI: 0.062, 0.505; ASH-CA: 0.309, 95% CI: 0.137, 0.673), ***IL6*** (ASH-CH: 0.274, 95% CI: 0.073, 0.838; ASH-CA: 0.232, 95% CI: 0.004, 1.097), ***IL10*** (ASH-CH: 0.193, 95% CI: 0.014, 0.381; ASH-CA: 0.215; 95% CI: 0.111, 1.406), ***UCP2*** (ASH-CH: 0.058, 95% CI: 0.113, 1.175; ASH-CA: 0.269, 95% CI: 0.208, 0.342), ***GPX3*** (ASH-CH: 0.241, 95% CI: 0.0634, 0.797; ASH-CA: 0.296, 95%CI: 0.102, 0.814), ***GPX4*** (ASH-CH: 0.207, 95% CI: 0.053, 0.816; ASH-CA: 0.239, 95% CI: 0.109, 0.508), and ***NFEL2*** (ASH-CH: 0.158, 95% CI: 0.028, 0.692; ASH-CA: 0.144, 95% CI: 0.052, 0.243).

Treatment of AT with the sera of participants who supplemented with ASH-CH or ASH-CA had no effect on the mRNA expression of *IL1β*, *IL-10*, *CD14*, *NFkB1*, *CCL2*, *MMP9*, *MMP14*, *APOE*, *IFNG*, *SOD1*, *SOD2*, *GPX1*, *CAT*, *iNOS.*


SKM cells treated with ASH-CH and ASH-CA had a higher expression of ***SREBF1*** (ASH-CH: 1.67, 95% CI: 0.385, 2.970; ASH-CA: 1.721, 95% CI: 1.110, 2.645) and a lower expression of ***FOXO1*** (ASH-CH: 0.564, 95% CI: 0.305, 0.855; ASH-CA: 0.924, 95% CI: 0.319, 2.585) (Figure 5). There was no effect of ASH-CH or ASH-CA on genes involved in inflammation/oxidative stress in SKM cells.

## 4. Discussion

We investigated for the first time the anti-inflammatory, antioxidant, and metabolic effects of Andrographis and ashwagandha in a physiologically relevant ex vivo model of human adipose tissue and skeletal muscle. ASH sera treatment altered the expression of fifteen metabolic, antioxidant defence, and inflammation genes in AT. AND treatment altered the expression of two AT inflammation/metabolism genes. ASH treatment also altered the expression of two genes in SKM.

### 4.1. Effect of Andrographis in AT 

*SERPINA3* expression was significantly lower in AT explants treated with AND-CH. Serpina3 is part of the superfamily of the serine protease inhibitor (serpin). The plasma SERPINA3 level increases during inflammation [42]. Seprina3 is highly expressed in AT, where it regulates preadipocyte differentiation, AT inflammation, and tumour necrosis factor alpha (TNF-α)-induced insulin resistance [43,44,45]. Knockdown of *Serpina3c* is associated with the inhibition of adipogenesis and adipocyte differentiation via an AKT-mediated decrease in the nuclear translocation of glycogen synthase kinase 3β (GSK3β) [43]. This is consistent with findings in humans, where increased *SERPINA3* expression is found in AT from obese individuals and individuals with cardiovascular disease [43,46,47]. Previous studies have reported an anti-adipogenic effect of Andrographis treatment [21,22]. However, we found no change in the expression of other regulators of AT differentiation (e.g., *PPARG*) or changes in *AKT1* or *SERPINE1* gene expression. It is therefore unclear whether the observed changes in SERPINA3 mRNA would lead to reduced *Serpina3c* protein expression and whether this would be necessary or sufficient to induce an anti-adipogenic effect in humans in vivo. This warrants further exploration. 

In our study, AT explants treated with AND-CH had lower expression of *IL18* compared to non-AND supplemented sera. IL-18 is a pro-inflammatory cytokine that is associated with obesity, insulin resistance, hypertension, dyslipidaemia, and cardiovascular disease [48]. AT levels of IL-18 have been found to be correlated with BMI, and serum IL-18 is significantly higher in people with T2D and metabolic syndrome [49,50,51]. Hyperglycaemia has been shown to increase IL-18 expression in adipocytes [52]. No other studies have reported an effect of Andrographis on IL-18 levels or expression. One study investigated the effect of andrographolide derivatives on the NF-κB pathway in LPS-stimulated THP-1 cells and found no effect of andrographolide derivatives on IL-18 mRNA expression [53]. The effects of Andrographis on IL-18 mRNA expression could be tissue-specific [54]. Crucially, our model is unlike previous in vitro research due to the use of human serum and human tissue, which better mimics in vivo physiology. Considering that IL-18 is a pleiotropic cytokine that is both a cause and a consequence of chronic metabolic inflammation, the clinical implications of the observed reduction in *IL-18* expression warrant further investigation in human clinical trials. 

Andrographis treatment has previously been shown to inhibit inflammation in other tissues via inhibition of a variety of signalling pathways and mediators (e.g., TNF-α, NF-κB, interleukin 6 [IL-6], monocyte chemoattractant protein-1 [MCP-1], interleukin 1 beta [IL1β]) in pre-clinical models of other inflammatory conditions [55]. In our study, treatment with Andrographis metabolites did not have an effect on other inflammatory markers. The discrepancy between previous research findings and the current study may be explained by (1) differences in phytochemical metabolism and immune function in rodents and humans, (2) in vitro models utilising the application of parent compounds rather than the circulating metabolites in the current study, (3) lack of the physiological relevance of animal-derived cell-based models, and (4) more pronounced effects in a high-inflammatory milieu compared to the model of subclinical inflammation in the current study [56,57,58,59].

### 4.2. Effect of Ashwagandha on AT and SKM

In this study, AT explants treated with ashwagandha metabolites had decreased mRNA expression of pro-inflammatory/pro-oxidant genes (*CCL5*, *IL6*) accompanied by a reduction in the expression of the anti-inflammatory/antioxidant gene mRNA (*IL10*, *UCP2*, *GPX3*, *GPX4*, *ADIPOQ*, *NFEL2*). 

*CCL5* encodes the C-C chemokine motif ligand 5, which is implicated in the development of obesity-associated AT inflammation and metabolic disturbances. In fact, CCL5 (along with CCL2) is a major recruiter of pro-inflammatory immune cells to AT [60]. CCL5 expression is increased in obese AT and directly correlated with macrophage inflammation. In fact, CCL5 mRNA expression has been shown to be correlated with M1 macrophage markers TNF-α and IL-6 [61]. Indeed, in our study, we found concomitant reductions in *IL-6* mRNA expression. The effects of ashwagandha treatment on AT inflammation have never previously been investigated. However, in microglial cells, ashwagandha extract (0.2% *v*/*v*) pre-treatment attenuated the LPS-stimulated production of IL-1β, IL-6, and TNF via downregulation of the NF-κB pathway [62]. In a double-blind RCT, 66 participants with type 2 diabetes supplemented with twice daily 250 mg or 500 mg ashwagandha extract for 12 weeks [32]. Ashwagandha supplementation produced a dose-dependent decrease in serum malondialdehyde levels (250 mg: −6.36%, 500 mg: −21.39%) and hs-CRP (250 mg: −41.22%, 500 mg: −57.71%), and a dose-dependent increase in serum glutathione (GSH) levels (250 mg: 14.72%, 500 mg: 31.48%) [32]. 

We found that AT explants treated with ASH-CH and ASH-CA displayed decreased mRNA expression of *UCP2*, *GPX3*, *GPX4*, and *NFEL2*. Our results contradict those reported in previous studies. First, in animal models of diet-induced obesity, Withaferin A supplementation reduced hepatic fat, restored hepatic insulin sensitivity, and increased superoxide dismutase (SOD), catalase (CAT), glutathione peroxidase (GPx), and GSH content in the liver [63,64]. Similarly, in obese mice, ashwagandha supplementation increased *UCP2* mRNA expression in SKM and promoted the browning of (subcutaneous) AT via increasing the expression of mitochondrial uncoupling protein 1 (UCP-1) [17,18]. *UCP2* upregulation protects cellular damage by decreasing intracellular ROS, while UCP1 upregulation provokes energy dissipation in the form of heat. It is notable that the mRNA expression of *NFEL2*, a transcription factor that regulates the defence against oxidative stress, was also reduced in the presence of ashwagandha metabolites. This suggests a decrease in the requirement for endogenous antioxidant defence in the presence of sera containing ASH metabolites. We hypothesise that a decrease in inflammation (as indicated by reduced *CCL5*, *IL6* expression) may contribute to this reduced requirement to transcribe antioxidant genes. This is speculative and requires extensive further exploration in human clinical trials. 

Treatment with ASH-supplemented sera also decreased the mRNA expression of the ‘anti-inflammatory’ adipokine adiponectin (*ADIPOQ*). Adiponectin is also a positive regulator of glucose homeostasis, lipid metabolism, and insulin sensitivity [65]. Ashwagandha metabolites have been shown to alter adipokine secretion in other studies. For example, in mice with diet-induced obesity, Withaferin A treatment significantly decreased plasma leptin concentrations but significantly restored leptin sensitivity, which in turn improved energy homeostasis and reduced bodyweight [16]. In the present study, we found no effect of ASH-supplemented sera on leptin mRNA expression in AT, and instead found a decrease in adiponectin expression. It is unclear how these changes would be reflected in plasma proteins as changes in mRNA expression do not necessarily translate into changes in circulating levels. Future investigations in clinical trials could help elucidate these effects.

Treatment with ashwagandha metabolites altered the expression of genes involved in AT energy metabolism, particularly fatty acid uptake and metabolism, and glucose metabolism. ASH-CH- and ASH-CA-treated AT explants had lower expression of *CD36*, *FABP4*, *CPT1B*, *CPT2*, *FASN*, *GSK3A*, and *NAMPT.* CD36 and FABP4 are involved in long chain fatty acid (LCFA) uptake, whereas CPT1B and CPT2 are involved in the transport of LCFA and Acyl-CoA transport into the mitochondria for β-oxidation. It has been shown that the lower uptake and oxidation of fatty acids is associated with reduced diet-induced inflammation and oxidative stress in AT but has no effect on adipogenesis [66]. These changes in energy metabolism genes might therefore be associated with a reduced overall inflammatory milieu, as suggested previously.

We propose that ashwagandha may—at least in part—exert its documented anti-adipogenic effects via the inhibition of *FASN*, which is known to promote adipogenesis and adipocyte differentiation [67]. Previous studies using 3T3-L1 preadipocytes suggested that the anti-adipogenic effects of ashwagandha were mediated by an increase in the mRNA expression of lipolytic genes (hormone-sensitive lipase (*HSL*) and adipose triglyceride lipase (*ATGL*)) and a downregulation of lipogenic genes (*SREBP1*) [68]. Another study also found that ashwagandha decreased lipid accumulation in 3T3-L1 preadipocytes, which was associated with decreased PPARγ protein expression [14]. We did not find any effect of ashwagandha-metabolite-containing sera on *PPARG* expression in AT (Appendix A). This could be due to the limited human applicability of 3T3-L1 adipocytes [69]. For example, studies comparing a primary human Simpson–Golabi–Behmel syndrome (SGBS) line and mouse 3T3-L1 lines found significant differences in the expression of marker genes [70]. The mouse and human line shared 295 adipocyte marker genes and displayed 445 and 860 unique marker genes in the human and mouse cell line, respectively. Mouse and human adipose tissue expansion and metabolic dysfunction are also significantly different [71]. For example, while human subcutaneous AT expands through hypertrophy and hyperplasia, mouse AT seems to expand only via hypertrophy [72]. This in turn leads to differences in the vascularisation of AT which results in different levels of tissue hypoxia, hypoxia-inducible factors, and resulting tissue inflammation [73,74].

In the present study, *NAMPT* expression was lower in treated AT. In mice, fat-specific *NAMPT* knockouts were resistant to high-fat-diet-induced obesity and had improved glucose tolerance [75]. It is unclear how these results translate to humans. Further investigations measuring lipid accumulation, adipocyte size, and protein expression are needed to better understand the effects of ashwagandha on lipid metabolism in AT. 

### 4.3. Limitations and Future Directions

We investigated the effects of Andrographis and ashwagandha on inflammation, pro-oxidant/anti-oxidant balance, and metabolism in order to screen these supplements for the potential to improve human cardiometabolic health. We recruited healthy volunteers and conducted ex vivo investigations in the SKMC and AT of healthy volunteers. This approach minimised inter-individual variability in background inflammation, essential for this screening study. However, our findings may not be applicable to a population with cardiometabolic disturbances [76,77,78]. It may be that a model employing sera and tissue from people with metabolic dysfunction and substantial systemic background inflammation may have revealed more substantial or different changes in mRNA expression. For adipose explants, had we recruited a population with cardiometabolic disturbances, it might have been questioned as to why we did not recruit a lean control group. This would then have necessitated additional human participants experiencing an adipose biopsy procedure for the purposes of an in vitro screening study—the ethical implications of this are evident. Our use of human adipose tissue, human sera, and myogenic cells from human skeletal muscle, while more physiologically relevant than many previous ex vivo approaches, carries its own limitations. First, inter-individual differences in adipose tissue physiology could have made it more difficult to detect the effects of the treatment [79]. The skeletal muscle cells were isolated from a single muscle biopsy from a healthy young male. This is preferable to a murine cell line, e.g., C2C12, but could none the less limit the applicability of our findings to our target population. 

We pooled the sera per supplement and timepoint; this resulted in measuring the ‘average’ response to supplementation. As such, this study exposed AT and SKM to (1) an ‘average’ concentration of ASH/AND metabolites that would be found in vivo and (2) accounted for the ‘average’ unexplored metabolite–serum interactions and any resulting effects on the overall inflammatory milieu. This maximises physiological relevance and is a major strength of this study.

While the supplementation period and dose were chosen based on the available literature, the lack of available pharmacokinetic data prevents the quantification of sera metabolites present in the sera at the time of collection. This remains a considerable challenge in the literature. A single study of ashwagandha pharmacokinetics has been described in humans. Plasma Withaferin A (n = 13) was measured after an acute ashwagandha dose (4800 mg containing 216 mg Withaferin A) and researchers could not detect one of ashwagandha’s known metabolites, Withaferin A, in any biological samples [80]. We hypothesise this could be due to the low sensitivity of the measurement technique and to the population (patients with advanced stage high-grade osteosarcoma). Our 1 h timepoint for acute supplementation was based on the best available information [81], but may not have captured peak metabolite concentrations. In fact, there is generally only a minor difference in effects between CH and CA conditions. Certain genes (e.g., the effect of ASH on *FABP4* in AT) were more markedly altered by the CA sera than by the CH alone, suggesting that the 1 h timepoint did capture, at least in part, the acute response to the supplement. There is a critical need for high-quality investigations of the pharmacokinetic profiles of ashwagandha and Andrographis to understand their metabolism and bioavailability. This is currently significantly constrained by a lack of exploratory work identifying circulating ashwagandha and Andrographis metabolites.

Our study exposed the SKMC and AT explant to a single ‘timepoint’ of metabolite concentration, whereas, in vivo, serum is a complex milieu that is constantly changing. No in vitro study design can accurately capture these dynamic changes in the proportion of each metabolite, the evolving concentration of each metabolite, and its time-dependent effects on other constituents in the circulation. Similarly, cells/tissues were treated for 24 h and mRNA expression was measured at a single timepoint. In humans, tissue-level changes in mRNA expression in response to an intervention depend on the timing of the biopsy. For example, following an exercise intervention, mRNA expression of *PPARα* is significantly altered 3, 24, 48, and 72 h, but not 9 h, after the intervention [82]. The use of a single timepoint may lead to an incomplete understanding of the molecular response to treatment. Our model nonetheless mimics the complexity of human physiology to a greater extent than past investigations and contributes strong evidence to justify comprehensive in vivo human trials. 

We have used confidence intervals, not *p*-values to describe our data. There are compelling reasons for this, which we would invite researchers to consider in handling similar data. A *p*-value characterises the evidence for or against the null hypothesis. It does not address the uncertainly in the size of effect or the issue of whether all plausible values are biologically important or are greater than a ‘zero effect’ threshold. Confidence intervals do this well, and values that do not cross the value of zero effect can be considered meaningful [40,41]. Further, exploratory/screening in vitro studies of gene and protein expression (including ours) typically employ limited numbers of experimental replicates. This is reasonable, given these studies are typically conducted to signpost future in vivo studies. In such scenarios, confidence intervals are superior to a binary *p*-value judgement regarding a response to treatment. The confidence interval approach allows the likely range of values to be ascertained, thus responsibly allowing researchers to pursue research avenues that show promise.

## 5. Conclusions

Our in vitro findings suggest that ashwagandha supplementation could alter antioxidant defence and inflammatory responses in vivo. Our work also suggests that fuel metabolism may be altered by ashwagandha in vivo. Andrographis supplementation could alter the expression of pro-inflammatory genes. Many of the circulating ashwagandha and Andrographis metabolites responsible for these effects have yet to be identified. How such metabolites act in concert in vivo is poorly understood. We suggest that an investigation of the in vivo effects of ashwagandha should be prioritised. Our data represent reasonable justification for a comprehensive randomised placebo-controlled trial of ashwagandha in a population with deteriorating cardiometabolic health. Such a trial should measure both circulating and tissue-level (adipose, skeletal muscle) markers of inflammation and energy metabolism. This would represent an important incremental step in determining the effect of ashwagandha on markers of human cardiometabolic health. 

## Figures and Tables

**Figure 1 nutrients-16-02291-f001:**
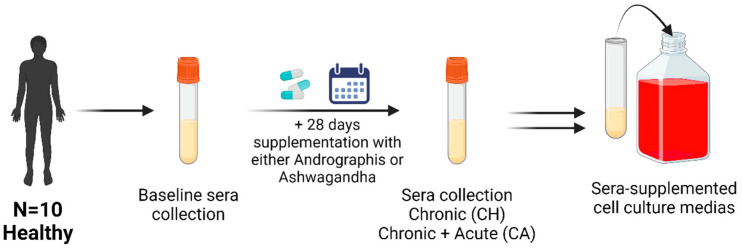
Outline of sera collection procedure. Healthy participants supplemented for 28 days with ashwagandha (N = 10, 1100 mg/day) and Andrographis (N = 10, 2000 mg/day). Sera were collected at baseline, 28 days post supplementation (chronic, CH), and +1 h after 28 days of supplementation and an acute dose of the supplement. The sera were pooled for all participants and used to obtain 6 sera-supplemented cell culture media: (1) ashwagandha baseline (ASH-BL), (2) chronic ashwagandha supplementation (ASH-CH), (3) chronic ashwagandha supplementation followed by acute dose of ashwagandha (ASH-CA), (4) Andrographis baseline (AND-BL), (5) chronic Andrographis supplementation (AND-CH), and (6) chronic Andrographis supplementation followed by an acute dose (AND-CA).

**Figure 2 nutrients-16-02291-f002:**
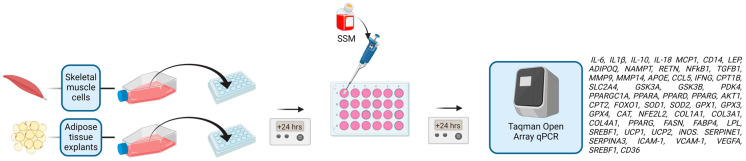
Procedure for the treatment of SKM cells and adipose tissue explants with the pooled sera of people who supplemented with ashwagandha or Andrographis (sera-supplemented media, SSM). Cells/tissues were treated with one of six SSM: (1) ashwagandha baseline (ASH-BL), (2) chronic ashwagandha supplementation (ASH-CH), (3) chronic ashwagandha supplementation followed by acute dose of ashwagandha (ASH-CA), (4) Andrographis baseline (AND-BL), (5) chronic Andrographis supplementation (AND-CH), and (6) chronic Andrographis supplementation followed by an acute dose (AND-CA). Experiments were performed in quadruplicate, with triplicate wells stimulated in each experiment. Polymerase chain reaction reactions were performed in triplicate.

**Figure 3 nutrients-16-02291-f003:**
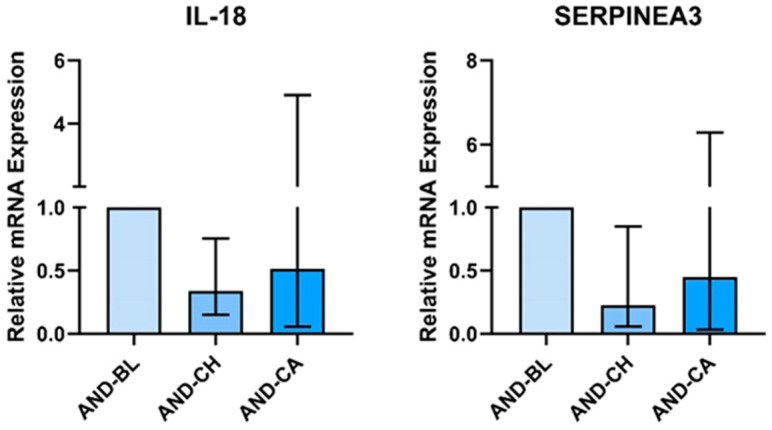
Adipose tissue explant mRNA expression is altered by treatment with sera of people who supplemented with Andrographis (AND). Sera were obtained following 28 days supplementation (chronic, (CH)) and following 28 days supplementation and an acute dose (+1 h) of AND (chronic + acute (CA)). Values are expressed relative to mRNA expression in samples treated with pre-supplementation baseline (BL) sera. Sera were obtained from 10 participants and pooled for adipose tissue treatment.

**Figure 4 nutrients-16-02291-f004:**
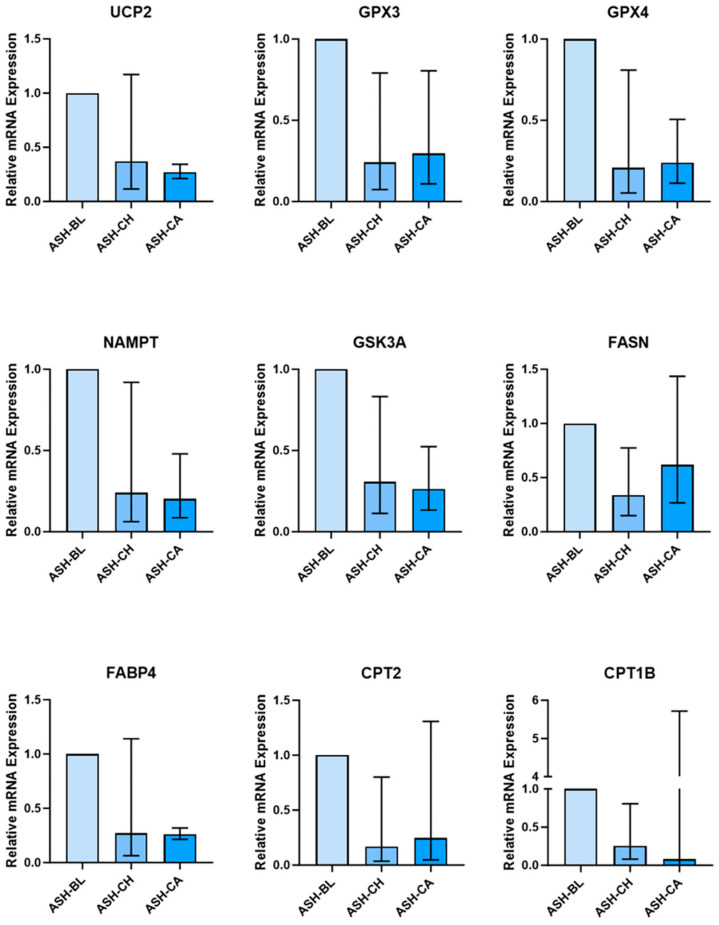
Adipose tissue explant mRNA expression is altered by treatment with sera of people who supplemented with ashwagandha (ASH). Sera were obtained following 28 days supplementation (chronic, (CH)) and following 28 days supplementation and an acute dose (+1 h) of ASH (chronic + acute (CA)). Values are expressed relative to mRNA expression in samples treated with pre-supplementation baseline (BL) sera. Sera were obtained from 10 participants and pooled for adipose tissue treatment.

**Figure 5 nutrients-16-02291-f005:**
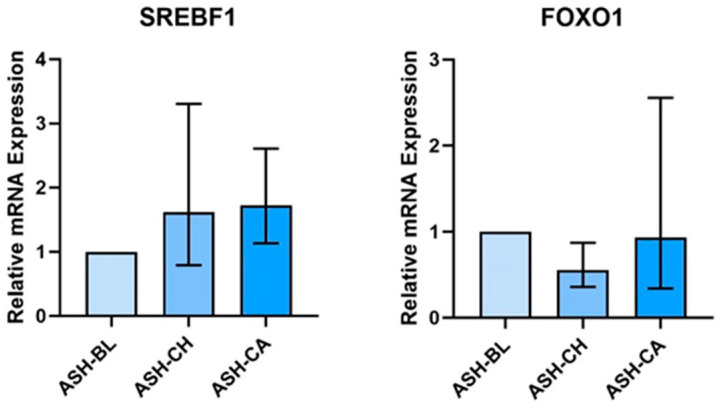
Skeletal muscle cells mRNA expression is altered by treatment with sera of people who supplemented with ashwagandha (ASH). Sera were obtained following 28 days supplementation (chronic, (CH)) and following 28 days supplementation and an acute dose (+1 h) of ASH (chronic + acute (CA)). Values are expressed relative to mRNA expression in samples treated with pre-supplementation baseline (BL) sera. Sera were obtained from 10 participants and pooled for skeletal muscle cell treatment.

**Table 1 nutrients-16-02291-t001:** Characteristics of participants recruited for sera collection. Participants supplemented with ashwagandha (N = 10, 1100 mg/day) or Andrographis (N = 10, 2000 mg/day) for 28 days.

	Ashwagandha (N = 10)	Andrographis (N = 10)
M/F	4/6	4/6
Age (years) (SD)	24.6 (4.0)	24.3 (3.9)
BMI (kg·m^2^) (SD)	23.0 (2.3)	21.6 (2.5)

**Table 2 nutrients-16-02291-t002:** Characteristics of adipose tissue donors (N = 7) recruited to investigate the effect of Andrographis and ashwagandha metabolites on adipose tissue metabolism and inflammation.

	Adipose Tissue Donors (N = 7)
M/F	4/3
Age (years) (SD)	25.3 (4.3)
BMI (kg·m^2^) (SD)	24.0 (2.8)

## Data Availability

The original contributions presented in the study are included in the article/Appendix A; further inquiries can be directed to the corresponding author.

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
