# Peer review of "Tissue-Level Effect of Andrographis and Ashwagandha Metabolites on Metabolic and Inflammatory Gene Expression in Skeletal Muscle and Adipose Tissue: An Ex Vivo/In Vitro Investigation"

_nutrients, 2024, doi:10.3390/nu16142291_

Round 1
Reviewer 1 Report
Comments and Suggestions for Authors
Please see the attached file.

Author Response
Reviewer 1
Thank you for your time and effort in reviewing this manuscript.
The manuscript does not indicate the approval number nor does it mention the registration number on an appropriate website (es. ISRCTN clinical trial registration).
Thank you for this late comment which was passed on by the handling editor. The study was approved by the University of Exeter Ethics Committee (ref: 22-02-02-A-06) and this is confirmed in the manuscript. We refer the reviewer to an excellent letter in Cell Metabolism which makes a compelling argument that physiology studies are not clinical trials and provides an explanation as to why registration and the requirement for compliance with ICMJE’s guidelines are unsuitable for experimental physiological studies (Richter et al. DOI:https://doi.org/10.1016/j.cmet.2024.04.005)
Comments:
- The term “v/v” should be italicized. Please modify it.
Thank you. Fixed, line 17 and 337.
- The abstract exceeds 200 words. Please reduce it to 200 words based on MDPI’s guidelines.
Our recount (Microsoft word, abstract text only) indicates that it was 200 words.
- The botanical names of the two plants cited in the study should be italicized within the text.
The full botanical names Withania somnifera and Andrographis paniculate were both italicized throughout. The foreshortened versions are not italicized for readability.
- Please ensure that all abbreviations are defined when first used and consider including a table of abbreviations (e.g., in line 42 “T2D” is not preceded by the extended name).
Thank you.
T2D corrected, line 42
BMI also defined, line 98
PES replaced with polyethersulfone (mentioned only once), line 130
HIV replaced with Human Immunodeficiency Virus (mentioned only once), line 147
TNF-α defined, line 301
MetS replace with metabolic syndrome (mentioned only once), line 317
IL-6, IL-1β, MCP-1 defined, line 330
Note: Genes mentioned as their standard gene names only e.g. NFEL2 are not defined for succinctness. This is important given the nature of this paper.
- Line 55: Why is the word “subcutaneous” included in parentheses? Ensure it is necessary.
Thank you for spotting this. We agree, changed.
- In Table 1, left column, there seems to be an error where standard deviations are indicated before the unit of measure (e.g., years) in parentheses. I believe they should be reversed.
Thank you for spotting this. We agree, changed in Table 1 and Table 2.
- The section titles should be modified according to MDPI guidelines (e.g., the “Participants” section should be changed to “2.1. Participants”). The same applies to the subsections.
Thank you. Errors in the methods and discussion have been corrected.
- The Materials and Methods section should include specifics regarding the ethics committee and the protocol approval number.
Thank you. This was added prior to the manuscript being returned to us by MDPI and we confirm that this can now be found in the relevant section.
- The statement that confidence intervals are judged more informative than statistical tests could be misinterpreted as avoiding rigorous statistical analysis. While it is valid to focus on confidence intervals, also consider mentioning any preliminary statistical tests conducted to support the findings. This adds an extra layer of validation to the results.
We are pleased that you agree with our approach as this is still sadly not common in the field despite good arguments in its favour. We have addressed this in full in response to your similar comment at the end of our response (see below)
Figures 4 and 5 are not cited in the text. Add the references.
Thank you for this, done. We have had to reverse the order of figures 4 and 5 so that they are presented in the order in which they are first mentioned in the text.
- The methods section is comprehensive but could be improved by providing more details on certain procedures. For example, I would suggest including more information on the rationale behind the chosen doses of ashwagandha and andrographis.
There is little precedent in the literature, but we agree that the rational for choice of doses needed clarification. This has been added to the methods section (line 116):
Empirical evidence regarding ashwagandha doses in humans in limited; this dose was chosen as 1) a similar dose (500 mg twice daily) has been shown to improve endothelial function in those with diabetes mellitus [32] and 2) this dose was in line with the manufacturer’s recommendations. Similarly, few human andrographis trials exist, with leaf extracts, rather than pure powdered leaf typically being used [33–35]. Therefore, the manufacturer’s recommended dose was used. This is an ecologically valid approach, as it is presumably the dose that is being consumed by existing andrographis users.
Again, I would suggest including statistical significance values (p-values) for the changes in gene expression to strengthen the validity of the findings.
Thank you for fundamentally agreeing with our approach to use confidence intervals here. We can appreciate the rationale for your suggestion to add p values, particularly the notion that it might create a perception of ‘hiding’ insights in the data given how prevalent this approach is in the field. However, on balance, we feel that presenting p values would undermine our strong belief that researchers should not continue to be beholden to what is not a good convention – i.e. that of presenting p values for exploratory in vitro research.
We do however agree that we need to provide a much stronger explanation for our approach in the text, in order to counteract this perception. We have done so in the limitations section and we hope that you agree with this approach (Line 481):
We have used confidence intervals, not p-values to describe our data. There are compelling reasons for this, which we would invite researchers to consider in handling similar data. A p-value characterises the evidence for or against the null hypothesis. It does not address the uncertainly in the size of effect or the issue of whether all plausible values are biologically important or are greater than a ‘zero effect’ threshold. Confidence intervals do this well, and values that do not cross the value of zero effect can be considered meaningful [40,41]. Further, exploratory/screening in vitro studies of gene and protein expression (including ours) typically employ limited numbers of experimental replicates. This is reasonable, given these studies are typically conducted to signpost future in vivo studies. In such scenarios, confidence intervals are superior to a binary p-value judgement regarding a response to treatment. The confidence interval approach allows the likely range of values to be ascertained, thus responsibly allowing researchers to pursue research avenues that show promise.
Reviewer 2 Report
Comments and Suggestions for Authors
Abstract. a fold change of 1 indicate no difference; why authors establish a cutoff of <1. It is usually considered of <-1.3 or >1.3 for gene expression. Authors must indicate significant differences, since these results are misleading.Authors indicated altered expressions that are inconsistent with the results described in the results sections, where no significant differences were observed.
Introduction. Authors must clearly justify the use of an ex vivo approach with healthy cells, when the introduction refer to obesity as the major concern. Furthermore, it is unclear why authors did not assess the effect in a clinical trial.
Materials & methods. Authors must justify the dosage of both materials in the chronic study. Why authors stablished 1 h for the acute consumption? Provide bioavailability studies that guarantee that 1 h after the consumption of these materials, the highest concentration of their major components are found in sera. Justify the dosage in the ex vivo study.
Results. No statistical analysis are included in Figures 3-5. If no significant difference was observed, authors must clearly state it in the caption. The results described in this section differ of those described in the abstract; since no differences were observed.
Discussion & conclusion. Authors discuss the beneficial effect of the treatments on the gene expression; however, these results are not observed in the figures. Therefore, all the discussion and conclusion section must be reconsidered
Author Response
Reviewer 2
Thank you for your time and effort in reviewing this manuscript.
Abstract. a fold change of 1 indicate no difference; why authors establish a cutoff of <1. It is usually considered of <-1.3 or >1.3 for gene expression. Authors must indicate significant differences, since these results are misleading. Authors indicated altered expressions that are inconsistent with the results described in the results sections, where no significant differences were observed.
There was indeed an error in the abstract thank you for noticing this. The text should have read geometric 95% CI for fold change > 1). This has been amended and may have been the source of some confusion for which we apologise. We will address your comments regarding the statistical analyses in full below (under ‘results’). However, in brief, we have used a legitimate (in both our view and in the view of the other reviewer) 95% confidence interval, rather than p-value approach to these analyses which we will clarify and explain further.
Introduction. Authors must clearly justify the use of an ex vivo approach with healthy cells, when the introduction refer to obesity as the major concern.
We acknowledge this as a weakness, although we feel that it is better addressed in the limitations section rather than the introduction.
We agree that ‘young healthy’ tissues and cells may respond differently than cells from individuals with chronic inflammation (e.g. in obesity). Using tissue and cells from older obese individuals would also be a legitimate approach. We were constrained in our access to these. We were also cognisant that if we were to use tissue from older/obese individuals then we would likely have to recruit more participants due to high degree of variability in their inflammatory profiles (O’Leary et al. doi: 10.1038/s41598-018-33840-x). Additionally, a legitimate comment might then have been to question why we did not recruit a lean control group, which would then have necessitated additional human participants experiencing an adipose biopsy procedure for the purposes of an in vitro screening study – the ethical implications of this are evident. Therefore, while these experiments in tissue from people exposed to chronic inflammation would have huge value, we consider the approach of using young healthy cells and tissues to be appropriate as a screening tool.
We had addressed these limitations in the relevant section of the manuscript, but this has now been expanded (Line 431):
We recruited healthy volunteers and conducted ex-vivo investigations in SKMC and AT of healthy volunteers. This approach minimised inter-individual variability in back-ground inflammation, essential for this screening study. However, our findings may not be applicable to a population with cardiometabolic disturbances [77-79]. It may be that a model employing sera and tissue from people with metabolic dysfunction and substantial systemic background inflammation may have revealed more substantial or different changes in mRNA expression. For adipose explants, had we recruited a population with cardiometabolic disturbances, it might have been questioned as to why we did not recruit a lean control group. This would then have necessitated additional hu-man participants experiencing an adipose biopsy procedure for the purposes of an in vitro screening study – the ethical implications of this are evident.
Furthermore, it is unclear why authors did not assess the effect in a clinical trial.
This is a natural next step but would have been orders of magnitude more expensive than our current approach. It would also have been burdensome for participants, which might not be considered ethical at this juncture given 1) the lack of in vitro evidence in human cells/tissues and 2) the proliferation of studies that have used herbal extracts rather than metabolites or sera of supplemented individuals. We have addressed this throughout the introduction and discussion. The findings of this study in our view now provide sufficient evidence to justify a clinical trial on the effects of ashwaganda, which we have recommended within the manuscript.
Materials & methods. Authors must justify the dosage of both materials in the chronic study. Why authors stablished 1 h for the acute consumption? Provide bioavailability studies that guarantee that 1 h after the consumption of these materials, the highest concentration of their major components are found in sera. Justify the dosage in the ex vivo study.
We interpret your comments as asking for justification for 1) the dosage (to be clear this is the same in both in vivo and ex vivo aspects of our work as they are the same sera) and 2) the 1h timepoint.
Dosage: The other reviewer also asked for clarification on this point. There is little precedent in the literature, but we agree that the rational for choice of doses needed clarification. This has been added to the methods section (line 116):
Empirical evidence regarding ashwagandha doses in humans in limited; this dose was chosen as 1) a similar dose (500 mg twice daily) has been shown to improve endothelial function in those with diabetes mellitus [32] and 2) this dose was in line with the manufacturer’s recommendations. Similarly, few human andrographis trials exist, with leaf extracts, rather than pure powdered leaf typically being used [33–35]. Therefore, the manufacturer’s recommended dose was used. This is an ecologically valid approach, as it is presumably the dose that is being consumed by existing andrographis users.
You have asked for a guarantee that the highest concentrations are found in the sera at 1h. This cannot be guaranteed, nor is it necessary to establish this for a screening study designed to provide evidence of biological effects to inform future studies. Rather, the many plasma metabolites derived from the supplements consumed merely need to be substantially enriched to demonstrate a biological effect. There is no necessity to use a timepoint that is the ‘absolute peak’. Indeed, this is not possible as our fundamental reason for applying sera is that ‘no in vitro study design can accurately capture these dynamic changes in the proportion of each metabolite, the evolving concentration of each metabolite, and its time-dependent effects on other constituents in the circulation’. Therefore, while we must seek some assurance that major metabolites have reached the circulation, this is the best that can reasonably be achieved. As already stated in the manuscript (line 457), the 1h timepoint was chosen based on the best available evidence of a substantial rise in circulating Andrographis metabolites at 1h (albeit with a peak occurring closer to 1.5 hours) (ref 82). There is little other pharmacokinetic data to guide such decision making. The only available pharmacokinetics clinical trial in humans was one in which plasma Withaferin A was measured after an acute Ashwagandha dose (up to 4,800 mg containing 216 mg Withaferin A). They could not detect Withaferin A in any biological samples (doi: 10.1016/j.jaim.2018.12.008). However, the assay (high performance liquid chromatography, HPLC) sensitivity was only 50 ng/mL. Additionally, in this study, the researchers did not attempt to measure compounds other than Withaferin A. In fact, conjugates and metabolites were not considered, but the chosen doses were safe and displayed biological activity in vivo. This study was also conducted in cancer patients receiving chemotherapy, which might have had a significant effect on the metabolism of Ashwagandha. Ultimately, our observation of biological effects provides reassurance that these doses and timings did alter the serum milieu. The next steps for this research, after establishing proof of biological effects from this study is to identify the main bioactive plasma metabolites and their pharmacokinetics.
Results. No statistical analysis are included in Figures 3-5. If no significant difference was observed, authors must clearly state it in the caption. The results described in this section differ of those described in the abstract; since no differences were observed.
We have deliberately not included statistical significance values and we believe that there are compelling reasons not to do this. We are happy to clarify for the reviewer why our approach is more appropriate in this instance and note that the other reviewer did agree with this being an appropriate primary approach to these data. Despite using p values in our previous published work, we now strongly believe that exploratory in vitro research should not be applying such tests to low ‘n’ measures of gene and protein expression. There are a number of compelling reasons for this in our and others views, which we would invite other researchers to also consider:
A p-value characterises the evidence for or against the null hypothesis. It does not address the uncertainly in the size of effect or the issue of whether all plausible values are biologically important (Halsey LG. 2019 The reign of the p-value is over: what alternative analyses could we employ to fill the power vacuum? Biol. Lett. 15: 20190174 http://dx.doi.org/10.1098/rsbl.2019.0174). Indeed, this is likely why the ‘convention’ of 1.3 fold changes in gene expression being considered meaningful emerged. It is a post hoc attempt to address the inherent weaknesses of analyses that rely on p values. We contend that this issue should be fixed ‘at source’ by employing confidence intervals, rather than by imposing this arbitrary threshold. Precedent is not a good enough reason for this practice to continue. Indeed, as noted in the manuscript, confidence intervals that do not include the value of zero effect (in this case ‘1’) can be assumed to be statistically significant (du Prel, J.B., Hommel, G., Röhrig, B., and Blettner, M., Confidence interval or p-value?: Part 4 of a series on evaluation of scientific publications. Dtsch Arztebl Int 2009, 106, 335-9. 10.3238/arztebl.2009.0335.
Further, exploratory/screening in vitro studies of gene and protein expression (including ours) typically employ limited experimental replicates. This is reasonable, given these studies are typically conducted to signpost future in vivo studies. Here in particular, confidence intervals are superior to a binary p-value judgement regarding a ‘yes/no’ response to treatment. The confidence interval approach allows the likely range of values to be ascertained, responsibly allowing researchers to pursue avenues that show promise.
We do however fully agree that we need to provide a much stronger explanation for our approach in the text, in order to counteract this perception. We have done so in the limitations section and we hope that you agree with this approach.
We have used confidence intervals, not p-values to describe our data. There are compelling reasons for this, which we would invite researchers to consider in handling similar data. A p-value characterises the evidence for or against the null hypothesis. It does not address the uncertainly in the size of effect or the issue of whether all plausible values are biologically important or are greater than a ‘zero effect’ threshold. Confidence intervals do this well, and values that do not cross the value of zero effect can be considered meaningful [40,41]. Further, exploratory/screening in vitro studies of gene and protein expression (including ours) typically employ limited numbers of experimental replicates. This is reasonable, given these studies are typically conducted to signpost future in vivo studies. In such scenarios, confidence intervals are superior to a binary p-value judgement regarding a response to treatment. The confidence interval approach allows the likely range of values to be ascertained, thus responsibly allowing researchers to pursue research avenues that show promise.
Discussion & conclusion. Authors discuss the beneficial effect of the treatments on the gene expression; however, these results are not observed in the figures. Therefore, all the discussion and conclusion section must be reconsidered
We disagree and we hope that the explanation above regarding the data handling provides some clarity and reassurance. Please note that we have studiously avoided using the term ‘significant’ in relation to our results precisely to prevent the impression that we are referring to p values. However, confidence intervals that do not include the value of zero effect (in this case ‘1’) can be assumed to be meaningful and therefore our discussion is appropriate.
Round 2
Reviewer 1 Report
Comments and Suggestions for Authors
I have reviewed the revisions made by the authors in response to the comments provided. The authors have made the necessary changes to improve the quality of the work that know is suitable for acceptance in its current form.